

# Development of nitrogen efficiency screening system in alfalfa (*Medicago sativa* L.) and analysis of alfalfa nitrogen efficiency types

Xiaojing Liu, Yajiao Zhao and Feng Hao

College of Pratacultural Science, Gansu Agricultural University, Lanzhou, Gansu Province, China

## ABSTRACT

Screening high nitrogen (N) efficiency crops is crucial to utilize resources rationally and reduce N losses. In this research, the biomass, morphological and N-related parameters of 28 alfalfa (*Medicago sativa* L.) cultivars were assessed at seedling stage. Then, we selected representative materials to compare the changes in stem-leaf dry weight (SDW), total root length (RL) and plant N accumulation (PNA) during whole period. Lastly, we analyzed the expressions of *NRT2* and *AMT1* genes of alfalfa cultivars. The correlation coefficients between SDW, PDW, RL, RV, SNA, RNA, and PNA were all in the range of 0.522~0.996. The coefficient of variations of SDW, PDW, RL, RV, SNA and PNA were all more than 20% under low and medium N levels. Though the comprehensive evaluation and cluster analysis, the comprehensive value of LW6010, Gannong NO.5, Longmu 806, Giant 2, Giant 601, Zhaodong, Crown were greater than 0.5 under low and medium N levels; the comprehensive value of Gannong NO.3, Gannong NO.4, Xinjiangdaye, Xinmu NO.1 were less than 0.5 under low N level, but were greater than 0.5 under medium N level. The comprehensive value of Gannong NO.7 Gannong NO.9, Longmu 801, Gongnong NO.3, Elite, Sadie 10, Giant 551 were greater than 0.5 under low N level, but were lesser than 0.5 under medium N level; and those of Longdong, Gannong NO.8, Gongnong NO.1, Reindee, Goldqueen, Weston, Tourists, Giant 6, Algonquin, Sadie 7 were lesser than 0.5 under low and medium N levels. Four N efficiency types of alfalfa cultivars were classified: (1) Very efficient; (2) Efficient; (3) Anti-efficient; and (4) Inefficient.The SDW, RL and PNA of LW6010 were higher than Longdong in each growth period. The expressions of *NRT2* and *AMT1* genes were highest for LW6010, and lowest for Longdong. So, N efficiency parameters assessed at seedling stage include: SDW, PDW, RL, RV, SNA and PNA. We developed new classification system of N efficiency types of alfalfa cultivars. It proved its effectiveness on 28 alfalfa in China.

# INTRODUCTION

Nitrogen (N), essential nutrient for all organisms, is the basic element of organic macromolecules. It has many functions in maintaining and regulating morphology, physiology and yield of crops. N fertilizer has been applied to maximize yield of crops;

Corresponding author
Xiaojing Liu, liuxj@gsau.edu.cn

however, most of the applied N fertilizer has been lost to air or leached into rivers and groundwater (*Guo et al., 2010*). It has been estimated that crops probably use less than half of the N fertilizer applied to soils (*Good, Shrawat & Muench, 2004*). Hence, it is imperative to develop high N efficiency cultivars to meet the needs for high yield and environmentally friendly agriculture (*Shi et al., 2010*). N efficiency refers to the comprehensive performance of crop growth, N absorption and utilization under different N environments (*Moll, Kamprath & Jackson, 1982*). Since *Smith (1934)* for the first time reported that there were significant differences in response of N in different maize (*Zea mays* L.) cultivars, the N efficiency among crops have become a research hotspot, and a large number of research projects been undertaken on a diverse range of crops. For instance, different wheat (*Triticum aestivum* L.) (*Pang et al., 2014*) and oilseed (*Brassica napus* L.) (*He et al., 2017*) cultivars also varied considerably in N use efficiency and N-related characteristics. Identification of crop cultivars with different N efficiency is important to improve economic and ecological benefits.

Previous studies quantified the performance of various plants such as Arabidopsis (*Arabidopsis thaliana* L.) (*Ikram et al., 2011*), rice (*Oryza.sativa* L.) (*Kessel, Schierholt & Becker, 2012*) and maize (*Wei et al., 2012*) under high-N and low-N environments. However, there has been little research on N efficiency screening, particularly on forage species. N efficiency screening systems have been developed and high N efficiency cultivars identified for cereal crops such as maize (*Chen et al., 2013*) and oil crops such as oilseed (*He et al., 2017*). Each N efficiency screening system consisted of different screening parameters due to different growth phenomenon, economic productions and harvest targets. Biomass was indispensable parameters for crops (*Balint & Rengel, 2008*). Root morphology directly affected the ability of crop to absorb N. At the same time, the N content and accumulation can be used to explain the N utilization ability of crop. Therefore, it is important to evaluate N efficiency of crop by using biomass, root morphology, N content and N accumulation.

N efficiency of crop is determined by external environment and internal genes. Crop uptake of exogenous N ($NH_4^+$-N and $NO_3^-$-N) mainly depends on ammonium N transporter (AMT) and nitrate N transporter (NRT), which are encoded by *AMT* and *NRT* gene families respectively (*Hu et al., 2015*). At the same time, *NRT2* and *AMT1* genes are high affinity transporters, and they were greatly affected by environmental N level (*Camane et al., 2012*). At present, the researches on the molecular mechanism of N use efficiency of crops should focus on rice. It is reported that the single base mutation of *OsNRT1.1B* gene is one of the important factors leading to the difference of N use efficiency among different rice cultivars (*Hu et al., 2015*). In our previous study, we found that compared other *NRT* and *AMT* family genes, *NRT2* and *AMT1* are most related to N content and N related enzyme activities etc. in alfalfa. Therefore, this study explored the expression of *NRT2* and *AMT1* genes of alfalfa under different N levels, providing scientific basis for screening N efficiency from molecular level, and providing method for rapid evaluation of N efficiency of alfalfa cultivars.

Alfalfa (*Medicago sativa* L.) is one of the earliest cultivated and most widely grown legume forages, and its protein content is much higher than that of cereal crops (*Shchebarskova, Kipaeva & Kadraliev, 2017*). Alfalfa can be used for biological N fixation,

but the fixed N cannot fully meet its growth needs (*Elgharably & Benes, 2021*). Therefore, the application of N fertilizer has become a necessary guarantee for high quality and efficient production of alfalfa. However, unreasonable N application not only reduced the N fixation capacity of alfalfa (*Roy et al., 2020*), but also caused serious environmental problems (*Guo et al., 2010*). Screening high N efficiency cultivars of alfalfa should be less fertilizer, less waste with higher N use efficiency and higher yield. In this study, different representative alfalfa cultivars were investigated, and N efficiency types of alfalfa linked to alfalfa were divided. It is expected that high N use efficiency, low N loss, and precision N fertilization can be achieved for alfalfa management through this development.

## MATERIALS & METHODS

### Experimental materials

A total of 28 alfalfa cultivars from various regions/countries such as China, America, Canada and Australia were selected as experimental materials (Table S1). The selected alfalfa cultivars had the rich characteristics, wide coverage and strong representation. At the same time, the selected types of alfalfa include imported cultivars and domestic certified cultivars. Twenty-eight alfalfa cultivars were selected from the temperate and warm temperate regions, northern temperate region, northern cold region, northwest inland irrigated agricultural region, Loess Plateau region and subtropical region.

### Treatments and design

#### Experiment 1—seedling screening experiment

This experiment was conducted in the temperature controlled greenhouse of Gansu Agricultural University, Lanzhou, China. In the temperature controlled greenhouse, light at 28 °C/14 h and darkness at 20 °C/10 h, light intensity was 260∼350 mol m$^{-2}$ s$^{-1}$, and relative humidity was 60%∼70%. It was 28 cultivars ×2 N levels = 56 treatments. There werethreereplications in each treatment (a total of 168 pots). The experiment was used nutrient solution sand culture method. Seed of the 28 alfalfa cultivars was disinfected, and then planted in pots (10 cm diameter, 13 cm height) filled with sand. When the alfalfa seedlings grew to three cm in height, 10 healthy seedlings were selected and kept in each sand pot with the rest removed by hand. 7 days after emergence of alfalfa, two levels of N nutrient solution (500 mL per pot) were added to the pots, and then rhizobium (*Sinorhizobium meliloti*, 12,531, provided by College of Pratacultural Science, Gansu Agricultural University, Lanzhou, China) liquid was inoculated (five mL per pot, OD$_{600}$ between 0.63 to 0.64) between the sand with the needles. The 2 N treatments were (1) the low N level (N21), this nutrient solution contained 21 mg L$^{-1}$ mixed N source of Ca(NO$_3$)$_2$ and (NH$_4$)$_2$SO$_4$, (2) the medium N level (N210, appropriate N levels of alfalfa), this nutrient solution contained 210 mg L$^{-1}$ mixed N source of Ca(NO$_3$)$_2$ and (NH$_4$)$_2$SO$_4$. Hoagland-Arnon solution as the basic nutrient solution was used in the low and medium N nutrient solutions, and the ration of NO$_3^-$-N: NH$_4^+$-N was 1:1 (*Hoagland & Arnon, 1950*; *Kou, 2011*). Sand was rinsed with distilled water and nutrient solution was replaced once a week. Distilled water (500 mL) was slowly added in each pot (prevent salt ion accumulation) in each time. Then pots with plants were rinsed with distilled water for
12 h, and then added nutrient solution (500 mL per pot). The method about nutrient solution sand culture method was same as Wang's research (*Wang et al., 2015*). Distilled water was supplemented every day to the location of the nutrient solution application for the first time. In this experiment, stem-leaf dry weight (SDW), root dry weight (RDW), whole plant dry weight (PDW), stem-leaf height (SH), total root length (RL), total root volume (RV), stem-leaf N content (SNC), root N content (RNC), whole plant N content (PNC), stem-leaf N accumulation (SNA), root N accumulation (RNA) and whole plant N accumulation (PNA) as the variables were used. The method as previously described in Zhao's research which is our team's preliminary research (*Zhao et al., 2020*).

### Experiment 2—the whole growth period (e.g., seedling stage, budding stage, flowering stage, pod stage and mature stage) verification experiment

This experiment was conducted using an outdoor rainout shelter at Gansu Agricultural University, Lanzhou, China. It was 4 cultivars ×2 N levels ×5 growth periods = 40 treatments. There were three replications in each treatment (a total of 120 pots). The experiment used nutrient solution sand culture method. Disinfected seed of four alfalfa cultivars (chosen cultivars were based on experiment 1) were planted in pots (32 cm diameter, 20 cm height) filled with sand. The four cultivars were: LW6010, Gannong NO.3, Gannong NO.7 and Longdong. There were two levels of N, the low N level (N content: 21 mg $L^{-1}$) and the medium N level (N content: 210 mg $L^{-1}$). The plant management was same as experiment 1. Stem-leaf and root were sampled at seedling, budding, flowering, pod and maturing stages, respectively. In this experiment, SDW, RL and PNA through the screened parameters of experiment 1 as the variables were used.

### Experiment 3—genes analysis experiment

This experiment was conducted in the temperature controlled greenhouse of Gansu Agricultural University, Lanzhou, China. It was 4 cultivars ×2 N levels = 8 treatments. There were three replications in each treatment (a total of 24 pots). The materials used for this experiment were identical to Experiment 2. The plant management, N treatments and sampled time were identical as in Experiment 1. When taking stem-leaf and root samples, the whole stem-leaf and root were collected and washed with water. Then they were cut into about one cm long segments, and approximately 100 mg material per sample pot was collected and put into liquid N quickly for RNA extraction. In this experiment, *NRT2* and *AMT1* genes as the variables were measured gene expressions.

## Morphological parameters

SH was measured with ruler on three randomly selected the tallest stem of plants per pot.

SDW, RDW and PDW were measured by collecting three plants from each pot. The samples were initially oven dried at 105 °C for 15 min and then at 65 °C until constant weight was achieved.

RV and RL were measured by collecting three plants from each pot. The entire root system was carefully removed by sliding it from its pot. The root system by spraying with water until it was almost free of sand particles. Each plant is separated by hand. Sieves of several mesh sizes (two mm, 500 mm, and 53 mm) were used to prevent loss of fine roots.

Prior to image analysis, roots were further cleaned by immersion in a water-filled basin, and any adhering particles were removed by hand. The stem was cut off and the root system washed, first by immersion in a water-filled container. Then roots were scanned with a scanner (EPSON Perfection V330, Epson Co., Suwa, Japan) and the total root length, root diameter, root surface area and root volume were obtained with the WinRHIZO Software (Regent Instruments Inc., Quebec, Canada).

## N Content

SNC, RNC and PNC were determined by Kjeldahl procedure after digestion in a mixture of concentrated $H_2SO_4$ and $H_2O_2$ (*Neide et al., 2003*). Powdered samples of stem-leaf, root and whole plant were digested in the Kjeldahl digestion flask by boiling with $H_2SO_4$-$H_2O_2$ until the mixture became clear. The digested liquid was filtered and volume. Ammonia was steam distilled from the digest to which NaOH solution was added. The distillate was collected in a conical flask containing HCl and red methyl indicator. The ammonia that was distilled into the receiving conical flask reacted with the acid and the excess acid in the flask was estimated by back titration against NaOH with color change from red to yellow (end point). Determinations were made on all reagents alone (blank determinations).

$$N(\%) = (V1 - V2) \times C \times 0.0140 \times 6.25/m \tag{1}$$

where *V1* is the amount of hydrochloric acid consumed during titration (mL), *V2* is the amount of hydrochloric acid consumed in the blank experiment (mL), *C* is the concentration of the standard solution of hydrochloric acid (mol/L), *m* is the sample mass (g), 6.25 is nitrogen conversion is the average coefficient of protein.

$$SNA \ (mg \ plant^{-1}) = SNC(\%) \times SDW \ (mg \ plant^{-1}) \tag{2}$$

$$RNA \ (mg \ plant^{-1}) = RNC(\%) \times RDW \ (mg \ plant^{-1}) \tag{3}$$

$$PNA(\%) = PNC \ (mg \ plant^{-1}) \times PDW \ (mg \ plant^{-1}) \tag{4}$$

$$N \ use \ efficiency = SDW/PNA \tag{5}$$

## N efficiency

N Efficiency represents the comprehensive performance of plant on the N concentration in the environment.

The membership function was used to comprehensively evaluate the N efficiency of alfalfa (*Zadeh, 1965*): $B_{mn} = (A_{mn}-A_{nmin})/(A_{nmax}-A_{nmin})$.

Where $B_{mn}$ isthe membership function value of m species with n index; $A_{mn}$ is the measured value of m species with n index; $A_{nmin}$ is the minimum value of all species with n index; $A_{nmax}$ is the maximum value of all species with n index; m is variety; n is index.

The weight is measured by objective weighting formula: $D_n = C_n / \Sigma C_n$.

Where $D_n$ is the weight; $C_n$ is the coefficient of variation.

N efficiency integrated value: N efficiency $= \sum(B_{mn} \times D_n)$.

## Quantitative real-time PCR analysis

Total RNA samples were isolated using TransZOL method. The concentration of the total RNA was determined by ultra-micro UV spectrophotometer (Quawell-Q5000, USA). A total of 5 ug of total RNA were used to synthesize cDNA by reverse transcriptase powerscript$^{TM}$ (Hiscript II Q Sellect RT SuperMix for qPCR (+ gDNA wiper) test kit; Vazyme; China) following the manufacturer's protocol. The cDNA samples were used as template to quantify the target gene expression levels (quantitative real-time PCR) by ChamQ SYBR Color qPCR Master Mix Test Kit (Vazyme, China) following the manufacturer's protocol.

According to the *Medicago truncatula* gene information (NCBI/GenBank accession NO. MTR 4g0057890 for *NRT2*; MRT 1g045550 for *AMT1*; MTR 8g098715 for 18s), we designed the primers for each gene, where *NRT2* is nitrate transporter gene, *AMT1* is ammonia transporter gene, and 18s is reference gene. The specific primers for the genes were: for *NRT2* forward: 5′-GGCAGCACCTTTAGTCCC-3′ and reverse: 5′-AATCTTCCTCGCATACCG-3′; for *AMT1* forward: 5′-TTCGTCACCCACCTACC-3′ and 5′-TTGTCACCGCCGTC CTC-3′; 18s forward: 5′- AAAGGAATTGACGGAAGGGC-3′ and 5′-CGCTCCA CCAACTAAGAACG-3′. Primer (synthesized by Shanghai bioengineering company, China) purity was of 99%. Primer concentration in PCR system was 1 μM.

Relative expression levels were calculated using the $2^{-\Delta\Delta CT}$ method.

$\Delta C_T(\text{test}) = C_T (\text{target, test}) - C_T(\text{ref, test})$

$\Delta C_T (\text{calibrator}) = C_T (\text{target, calibrator}) - C_T (\text{ref, calibrator})$

$\Delta\Delta C_T = \Delta C_T (\text{test}) - \Delta C_T(\text{calibrator})$

$2^{-\Delta\Delta CT}$ = Normalized expression ratio (relative expression level).

## Statistical analysis

All statistical analyses were performed using the Statistical Package for Windows (version 13.00; SPSS, Chicago, IL, USA). The suitable parameters of N efficiency screening can be screened out though the coefficient of variation of each parameter among different cultivars and the correlation between each parameter. The coefficient of variation can reflect the differences between alfalfa cultivars. The correlation between parameters can reflect the importance of parameters. Correlations between the investigated parameters were established by calculating simple pair-wise correlation coefficients. The comprehensive analysis of N efficiency of alfalfa cultivars were obtained by membership function and compounding operation. The membership function method can eliminate the inherent biological and genetic differences between different alfalfa cultivars, as well as the differences between dimensions and orders of different parameters (traits). The weight values of evaluation parameters were obtained by empirical weighting. Twenty-eight alfalfa cultivars were classified by cluster analysis under different N levels. And then, four types of N efficiency were determined though the results of cluster analysis and compounding value of N efficiency. The experimental data were analyzed using ANOVA and Tukey test at a significance level of $p < 0.05$.

**Table 1** Correlation of different alfalfa parameters under low nitrogen level (N21) at the seedling stage.

| | SDW | RDW | PDW | SH | RL | RV | SNC | RNC | PNC | SNA | RNA | PNA |
|---|---|---|---|---|---|---|---|---|---|---|---|---|
| SDW | 1 | | | | | | | | | | | |
| RDW | 0.416* | 1 | | | | | | | | | | |
| PDW | 0.907** | 0.758** | 1 | | | | | | | | | |
| SH | 0.385* | −0.186 | 0.190 | 1 | | | | | | | | |
| RL | 0.642** | 0.358 | 0.626** | 0.422* | 1 | | | | | | | |
| RV | 0.950** | 0.447* | 0.890** | 0.371 | 0.695** | 1 | | | | | | |
| SNC | 0.352 | −0.001 | 0.259 | 0.604** | 0.443* | 0.371 | 1 | | | | | |
| RNC | 0.275 | −0.100 | 0.141 | 0.454* | 0.328 | 0.251 | 0.392* | 1 | | | | |
| PNC | 0.416* | −0.091 | 0.257 | 0.652** | 0.477* | 0.409* | 0.888** | 0.761** | 1 | | | |
| SNA | 0.917** | 0.307 | 0.802** | 0.563** | 0.684** | 0.885** | 0.688** | 0.362 | 0.685** | 1 | | |
| RNA | 0.549** | 0.833** | 0.775** | 0.094 | 0.522** | 0.568** | 0.194 | 0.447* | 0.321 | 0.486** | 1 | |
| PNA | 0.902** | 0.552** | 0.903** | 0.460* | 0.716** | 0.885** | 0.594** | 0.449* | 0.642** | 0.941** | 0.752** | 1 |

**Notes.**

Note: SDW, RDW, PDW, SH, RL, RV, SNC, RNC, PNC, SNA, RNA and PNA represent stem-leaf dry weight, root dry weight, whole plant dry weigh, plant height, total root length, total root volume, stem-leaf nitrogen content, root nitrogen content, whole plant nitrogen content, stem-leaf nitrogen accumulation, root nitrogen accumulation and whole plant nitrogen accumulation correspondingly.

Asterisks (* and **) represent significance at the 0.05 and 0.01 two tailed levels.

## RESULTS

### N efficiency screening system at seedling stage

SDW, PDW, RL, RV, SNA, RNA, and PNA of alfalfa under N21 and N210 in seedling stage had different phenomenon.

Under N21, there was a significant positive correlation between SDW, PDW, RL, RV, SNA, RNA, and PNA ($p < 0.05$), and the range of correlation coefficient was from 0.522 to 0.950 (Table 1). Under N210, SDW, PDW, SH, RL, RV, SNA and PNA showed significant positive correlation ($p < 0.05$), the correlation coefficient being from 0.538 to 0.996 (Table 2). SDW, PDW, RL, RV, SNA and PNA showed significant positive correlation both under N21 and N210 ($p < 0.05$). The coefficient of variations (CVs) of SDW, RDW, PDW, RL, RV, SNA, RNA and PNA were all more than 20% under N21 and N210 (Table 3). Under different N levels, parameters that were significant correlations and high CVs could reflect the differences of N in different alfalfa cultivars. SDW, PDW, RL, RV, SNA and PNA could reflect the differences among cultivars of alfalfa.

Under N21 and N210, the weight value of SNA was the highest (0.208 and 0.201 for N21 and N210 respectively), and that of PDW was the lowest (0.143 and 0.136 for N21 and N210) (Table 4). Under N21, the maximum comprehensive parameter of N efficiency was 0.998 (LW6010) and under N21 was 0.015 (Longdong). The comprehensive parameters of N efficiency of three cultivars were more than 0.8, 14 cultivars were between 0.4 and 0.8 and 11 cultivars were below 0.4 (Table 5). Under N210, the highest comprehensive parameters of N efficiency was 0.980 (LW6010) and lowest was 0.068 (Gannong NO.8). There were four cultivars with comprehensive parameters of N efficiency more than 0.8, 9 cultivars between 0.4 and 0.8, and 15 cultivars below 0.4. Collectively the comprehensive parameters of N efficiency of three cultivars (LW6010, Giant 601 and Gannong NO.5) were

**Table 2  Correlation of different alfalfa parameters under medium nitrogen level (N210) at the seedling stage.**

|     | SDW | RDW | PDW | SH | RL | RV | SNC | RNC | PNC | SNA | RNA | PNA |
|-----|-----|-----|-----|-----|-----|-----|-----|-----|-----|-----|-----|-----|
| SDW | 1 | | | | | | | | | | | |
| RDW | 0.356 | 1 | | | | | | | | | | |
| PDW | 0.996** | 0.435* | 1 | | | | | | | | | |
| SH | 0.538** | 0.160 | 0.539** | 1 | | | | | | | | |
| RL | 0.708** | 0.454* | 0.728** | 0.549** | 1 | | | | | | | |
| RV | 0.799** | 0.394* | 0.812** | 0.610** | 0.825** | 1 | | | | | | |
| SNC | 0.342 | 0.175 | 0.344 | 0.370 | 0.540** | 0.426* | 1 | | | | | |
| RNC | 0.235 | 0.089 | 0.233 | 0.129 | 0.260 | 0.307 | 0.499** | 1 | | | | |
| PNC | 0.336 | 0.193 | 0.340 | 0.363 | 0.536** | 0.433* | 0.997** | 0.559** | 1 | | | |
| SNA | 0.869** | 0.349 | 0.868** | 0.573** | 0.766** | 0.791** | 0.750** | 0.444* | 0.748** | 1 | | |
| RNA | 0.451* | 0.903** | 0.520** | 0.272 | 0.560** | 0.539** | 0.375* | 0.470* | 0.412* | 0.529** | 1 | |
| PNA | 0.867** | 0.408* | 0.872** | 0.570** | 0.775** | 0.798** | 0.746** | 0.463* | 0.747** | 0.998** | 0.586** | 1 |

**Notes.**
See the note to Table 1.

**Table 3  SDW, RDW, PDW, SH, RL, RV, SNC, RNC, PNC, SNA, RNA and PNA ranges, averages and coefficient of variations (CVs) of alfalfa cultivars at the seedling stage under different nitrogen levels.**

| Index | Low nitrogen level (N21) | | | N210 | | |
|-------|-------|---------|--------|-------|---------|--------|
|       | Range | Average | CV (%) | Range | Average | CV (%) |
| SDW (g) | 0.026∼0.058 | 0.042 | 22.3 | 0.061∼0.125 | 0.097 | 20.8 |
| RDW (g) | 0.013∼0.035 | 0.024 | 24.3 | 0.006∼0.012 | 0.009 | 21.2 |
| PDW (g) | 0.039∼0.084 | 0.066 | 20 | 0.071∼0.136 | 0.106 | 20 |
| SH (cm) | 4.93∼9.67 | 7.38 | 15.2 | 12.27∼18.03 | 15.2 | 10.1 |
| RL (cm plant$^{-1}$) | 48.31∼98.97 | 68.49 | 20 | 30.11∼55.72 | 42.39 | 20.1 |
| RV (cm$^3$) | 27.67∼87.33 | 56.9 | 23.6 | 23.00∼55.33 | 36.82 | 28.4 |
| SNC (%) | 1.12∼1.64 | 1.35 | 11.5 | 2.11∼3.58 | 2.88 | 14.8 |
| RNC (%) | 0.92∼1.74 | 1.26 | 14 | 2.64∼3.69 | 3.25 | 9.4 |
| PNC (%) | 1.07∼1.55 | 1.32 | 10.8 | 2.15∼3.58 | 2.91 | 13.7 |
| SNA (mg plant$^{-1}$) | 0.34∼0.90 | 0.57 | 29.1 | 1.34∼4.45 | 2.82 | 29.4 |
| RNA (mg plant$^{-1}$) | 0.17∼0.52 | 0.3 | 27.4 | 0.20∼0.46 | 0.29 | 24.6 |
| PNA (mg plant$^{-1}$) | 0.51∼1.27 | 0.87 | 25.1 | 1.68∼4.80 | 3.1 | 27.9 |
| NUE (kg kg$^{-1}$) | 64.4∼93.2 | 76.83 | 10.8 | 27.9∼46.4 | 35.03 | 14.9 |

**Notes.**
See the note to Table 1.

above 0.8, 6 cultivars (longdong, Weston, Giant 6, Gongnong NO.1, Tourists and Sadi 7) were below 0.4. Alfalfa cultivars can be divided into different N efficiency types through the cluster analysis results and comprehensive values of different alfalfa cultivars under N21 and N210 (Figs. 1 and 2). The comprehensive value of LW6010, Gannong NO.5, Longmu 806, Giant 2, Giant 601, Crown, and Zhaodong were greater than 0.5 under N21 and N210. The comprehensive value of Gannong NO.3, Xinmu NO.1, Xinjiangdaye and Gannong NO.4 were lesser than 0.5 under N21, but were greater than 0.5 under N210. The comprehensive value of Sadie 10, Giant 551, Elite, Longmu 801, Gannong NO.3, Gannong

**Table 4  Weight values of SDW, PDW, RL, RV, SNA and PNA for 28 alfalfa cultivars under different nitrogen levels at the seedling stage.**

| Weight value | SDW (g) | PDW (g) | RL (cm plant$^{-1}$) | RV (cm$^3$) | SNA (mg plant$^{-1}$) | PNA (mg plant$^{-1}$) |
|---|---|---|---|---|---|---|
| N21 | 0.159 | 0.143 | 0.143 | 0.169 | 0.208 | 0.179 |
| N210 | 0.142 | 0.136 | 0.137 | 0.193 | 0.201 | 0.191 |

Notes.
See the note to Table 1.

**Table 5  Comprehensive N efficiency of alfalfa cultivars under low (N21) and medium (N210) nitrogen levels at the seedling stage.**

| Cultivars | Comprehensive N efficiency | | Cultivars | Comprehensive N efficiency | |
|---|---|---|---|---|---|
| | N21 | N210 | | N21 | N210 |
| Gannong NO.3 | 0.214 ± 0.074gk | 0.711 ± 0.106abcd | LW6010 | 0.910 ± 0.073b | 0.967 ± 0.044ab |
| Gannong NO.4 | 0.384 ± 0.059gh | 0.769 ± 0.053abcd | Reindeer | 0.426 ± 0.043 g | 0.276 ± 0.077abcd |
| Gannong NO.5 | 0.866 ± 0.071b | 0.840 ± 0.044abc | Crown | 0.535 ± 0.076ef | 0.798 ± 0.044abc |
| Gannong NO.7 | 0.593 ± 0.130de | 0.267 ± 0.045abcd | Goldqueen | 0.214 ± 0.043gk | 0.468 ± 0.077abcd |
| Gannong NO.8 | 0.135 ± 0.071l | 0.068 ± 0.049d | Giant 551 | 0.617 ± 0.047d | 0.339 ± 0.039abcd |
| Gannong NO.9 | 0.711 ± 0.052c | 0.286 ± 0.040abcd | Giant 601 | 0.998 ± 0.120a | 0.980 ± 0.083a |
| Longdong | 0.015 ± 0.072m | 0.169 ± 0.007cd | Giant 6 | 0.331 ± 0.036hi | 0.285 ± 0.044abcd |
| Xinjiangdaye | 0.296 ± 0.035i | 0.773 ± 0.120abcd | Giant 2 | 0.548 ± 0.123def | 0.747 ± 0.062abcd |
| Xinmu NO.1 | 0.438 ± 0.062 g | 0.714 ± 0.027abcd | Sadie 7 | 0.179 ± 0.054kl | 0.183 ± 0.048cd |
| Longmu 806 | 0.510 ± 0.092f | 0.814 ± 0.081abc | Sadie 10 | 0.579 ± 0.055def | 0.420 ± 0.051abcd |
| Longmu 801 | 0.754 ± 0.088c | 0.346 ± 0.059abcd | Tourists | 0.033 ± 0.043m | 0.255 ± 0.047bcd |
| Gongnong NO.1 | 0.281 ± 0.047ig | 0.247 ± 0.102bcd | Elite | 0.770 ± 0.039c | 0.326 ± 0.031abcd |
| Gongnong NO.3 | 0.604 ± 0.006de | 0.166 ± 0.085cd | Weston | 0.399 ± 0.084gh | 0.216 ± 0.084cd |
| Zhaodong | 0.558 ± 0.094def | 0.766 ± 0.040abcd | Algonquin | 0.400 ± 0.078gh | 0.329 ± 0.084abcd |

NO.7, Gannong NO.9 were greater than 0.5 under N21, but were lesser than 0.5 under N210. The comprehensive value of Longdong, Gannong NO.8, Reindeer, Goldqueen, Weston, Tourists, Giant 6, Gongnong NO.1, Algonquin and Sadie 7 were lesser than 0.5 under N21 and N210. So, 28 alfalfa cultivars could be classified into four types: (1) Very efficient, *e.g.*, LW6010; (2) Efficient, *e.g.*, Gannong NO.3; (3) Anti-efficient, *e.g.*, Gannong NO.7; and (4) Inefficient, *e.g.*, Longdong.

**N efficiency of different alfalfa types during the whole growth period**

SDW of the four alfalfa cultivars (cv. LW6010, Gannong NO.3, Gannong NO.7 and Longdong) increased with growth, which were greater under N210 than under N21 over the whole growth period (Fig. 3A). Under N210, the SDW of LW6010 and Gannong NO.3 were significantly higher than that of Gannong NO.7 and Longdong ($p < 0.05$). The SDW of LW6010 during pod stage was significantly higher than that of Gannong NO.3 ($p < 0.05$), but no significant difference was detected at other growth stages ($p > 0.05$). Under N21, the SDW of LW6010 and Gannong NO.7 were significantly higher than that of Gannong NO.3 and Longdong ($p < 0.05$); however, there was no significant difference ($p > 0.05$) between Gannong NO.7 and Longdong. Under both N levels, the SDW of LW6010 was remarkably greater than that of Longdong ($p < 0.05$).

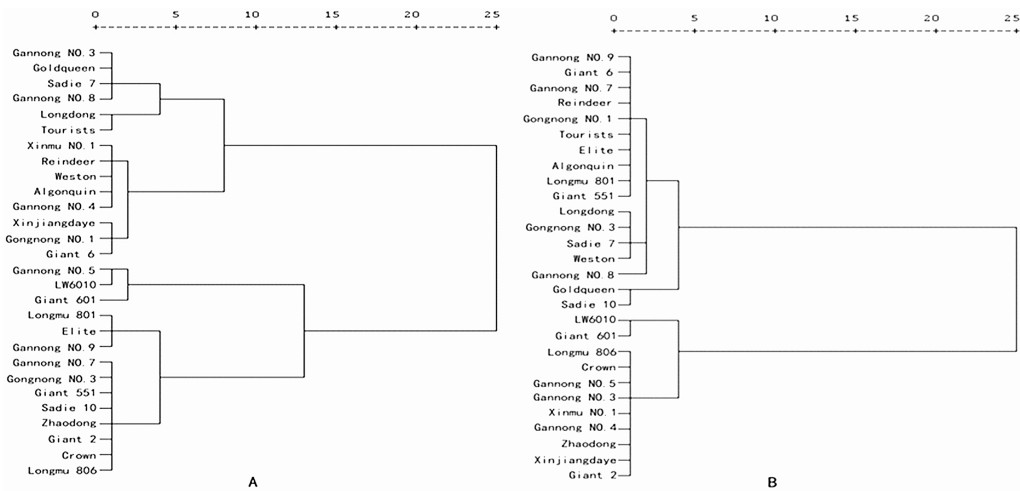

**Figure 1** Cluster analysis of nitrogen efficiency of 28 alfalfa cultivars under (A) low nitrogen level (N21), and (B) medium nitrogen level (N210).

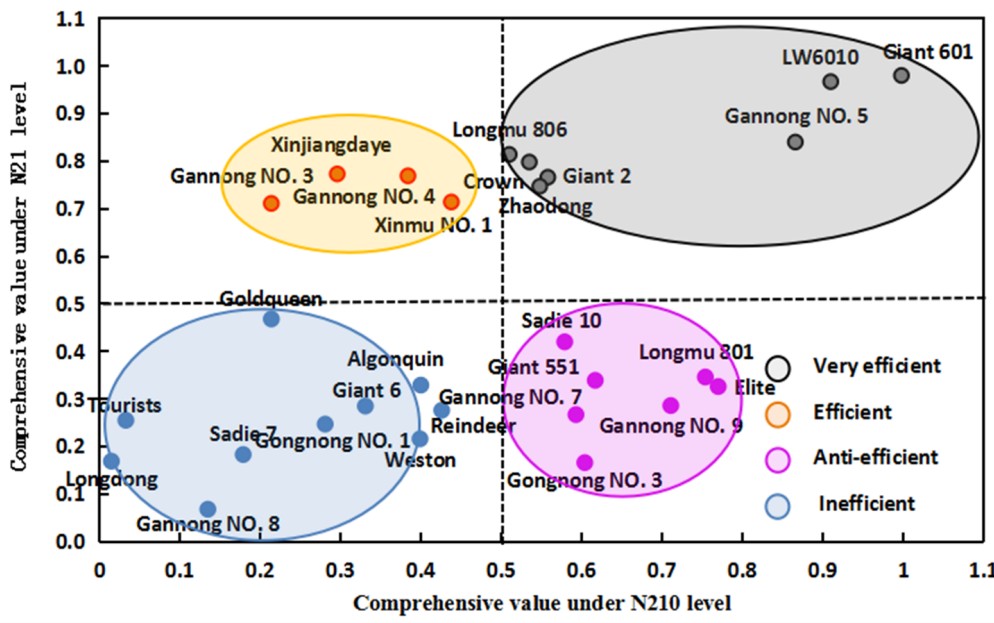

**Figure 2** Comprehensive values of 28 alfalfa cultivars under low nitrogen level (N21) and medium nitrogen level (N210).

The RL of the four alfalfa cultivars increased rapidly from seedling to flowering stages, but slowed down from flowering to maturity (Fig. 3B). From seedling to budding stage, the RL was lower under N21 than N210. Under N210, the RL of LW6010 and Gannong NO.3 were higher than that of Gannong NO.7 and Longdong for all growth periods. The RL of LW6010 was not different with Gannong NO.3 ($p > 0.05$) except at the pod stage.

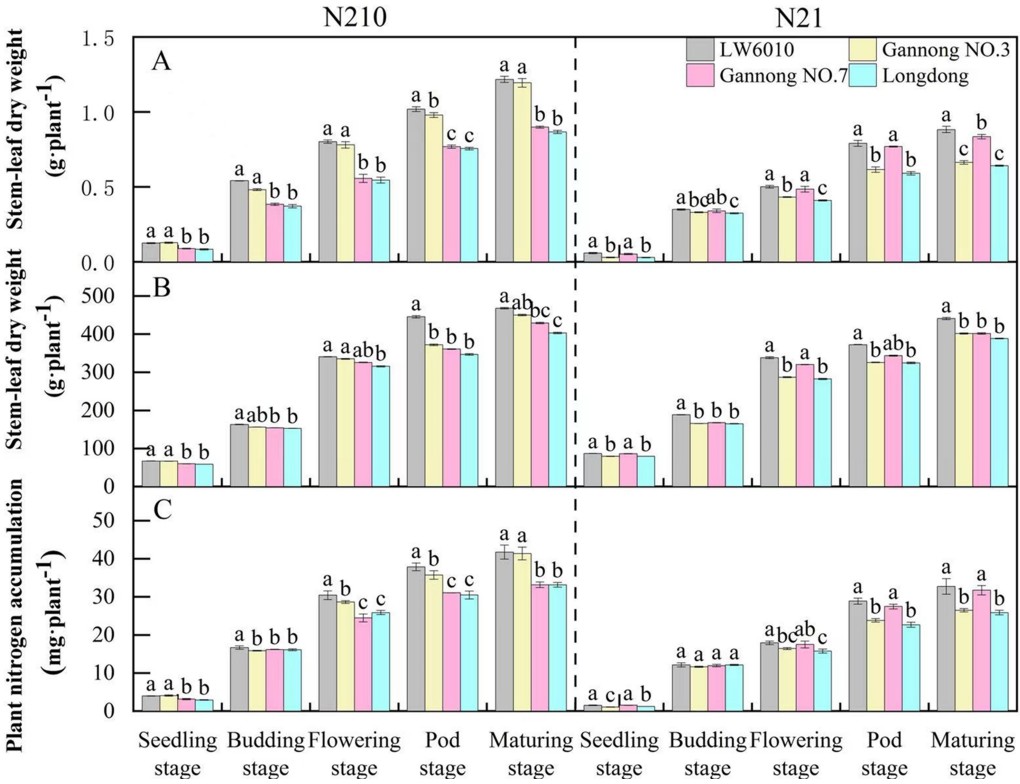

**Figure 3** **Effects of nitrogen levels on stem-leaf dry weight (A), root length (B) and plant nitrogen accumulation (C) for different nitrogen efficiency cultivars.** N21 and N210 represent low nitrogen level and medium nitrogen level. LSD means with different letters in the same stage under different nitrogen levels are significantly different at $p < 0.05$.

Under N21, the RL of LW6010 and Gannong NO.7 were greater than those of Gannong NO.3 and Longdong for all growth periods.

The PNA of the four alfalfa cultivars was greater under N210 than N21 over the whole growth period (Fig. 3C). The smallest PNAs of the four alfalfa cultivars all appeared at the flowering stage. The PNA of LW6010 was significantly higher than that of other cultivars ($p < 0.05$) from budding to pod stages under N210. The PNA of LW6010 and Gannong NO.3 was greater than that of Gannong NO.7 and Longdong under N210. Under N21, the PNA of LW6010 and Gannong NO.7 was greater than that of Gannong NO.3 and Longdong ($p < 0.05$).

## Response of N metabolism genes in seedling stage

As in Fig. 4, the relative expressions of *NRT2* and *AMT1* genes were higher in root than in stem-leaf, and their expressions under medium N level were higher than those under N21. Under N210, the relative expression of *NRT2* genes in stem-leaf of LW6010 was significantly higher than that of the other three cultivars ($p < 0.05$), and that of Longdong was significantly lower than that of the other cultivars ($p < 0.05$). Under N21, the relative expressions of *NRT2* in root were significantly higher for LW6010 than for the other

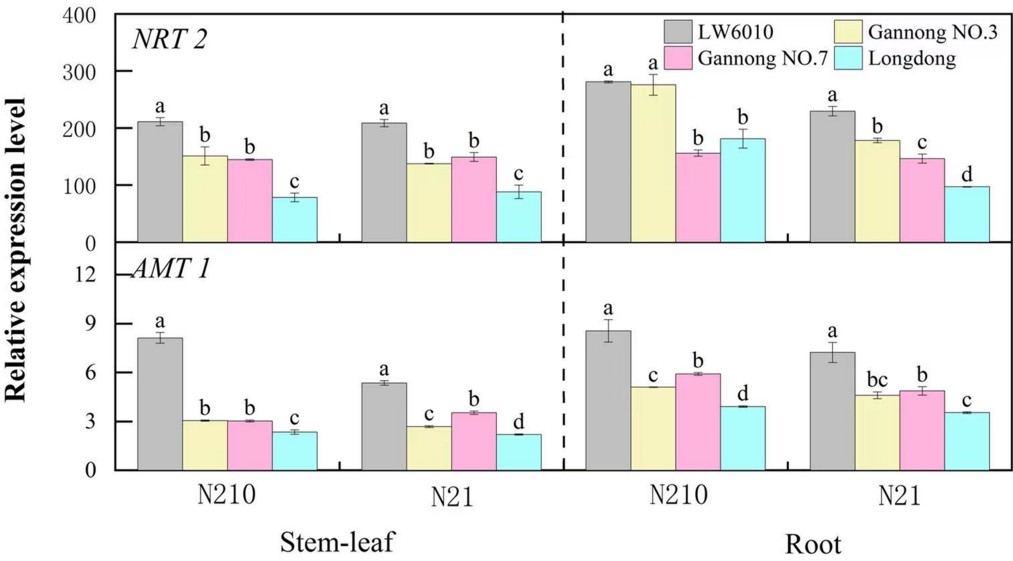

**Figure 4 Difference in *NRT2* and *AMT1* expression in stem-leaf and root for four representative alfalfa cultivars at the seedling stage.** N21 and N210 represent low nitrogen level and medium nitrogen level. LSD means with different letters under the same nitrogen level with different parts of plant are significantly different at $p < 0.05$.

cultivars ($p < 0.05$), and that of Longdong was the lowest. Under N210, the relative expression of *AMT1* in stem-leaf of LW6010 was significantly higher than that of the other three cultivars ($p < 0.05$), and that of Longdong were the lowest ($p < 0.05$). Under N21, there were significant differences ($p < 0.05$) in the relative expressions of *AMT1* in stem-leaf between the four alfalfa cultivars.

## DISCUSSION

In this study, 28 alfalfa cultivars that were selected had different responses to the same N level and different sensitivities to different N levels. They also had different expressions of N-relative genes, leading to differences in N efficiency. In our previous study, we also found that Gannong NO.3 and Longdong showed significant difference in SDW and SNA when applied with N fertilizer (*Liu et al., 2013*). It has been reported that other crops such as wheat (*Gaju et al., 2011*), rice (*Ju et al., 2015*) and canola (*Brassica napus* L.) (*Balint, Rengel & Allen, 2008*) also had different N efficiency in terms of yield, morphological characteristics, and N use efficiency components.

### The screening parameters of N efficiency of alfalfa were determined

The screening period is also an important factor in the screening system (*Xu et al., 2016*). Currently, N efficiency screening usually is performed during whole growth period (*Singh et al., 1998*) or at seedling stage (*Feng et al., 2014*). Screening during whole growth period has high accuracy, especially for cereal crops. However, it is not suitable for large scale screening because it is labor-intensive, time consuming, and hard to control the influencing factors. So screening at seedling stage is accepted as a substitute. Since prescreening of germplasm

at seedling stage could be an efficient tool for accelerating the selection process (*Schum & Jansen, 2014*), this approach may be helpful for initial screening of germplasm and identification of divergently responding genotypes. Screening of cultivars with different N efficiency at the seedling stage was not only used in cereal crops, such as rice (*Feng et al., 2014*), maize (*Li et al., 2014*), foxtail millet (*Chen et al., 2016*), but also in harvesting vegetable crops, such as rapeseed (*Brassica napus* L.) (*Zou et al., 2017*), spinach (*Spinacia oleracea* L.) (*Liu et al., 2012*), and cucumber (*Cucumis sativus* L.) (*Zhao et al., 2015*). N efficiency screening in alfalfa cultivars at seedling stage may also be an efficient screening technique. The most important factors of N efficiency screening system in crops mainly include screening parameters, screening period and screening method. At present, the parameters in N efficiency screening systems are focused on biomass, morphological and physiological characteristics and N-related characteristics (*Wang et al., 2014*). In this study, the N efficiency screening parameters at seedling stage were SDW, PDW, RL, RV, SNA and PNA, which were screened from growth-related and N-related parameters (their CVs were more than 20% and they were significantly positively correlated). SDW and PDW can be critical components for crop yield. *Balint & Rengel (2008)* also used dry weight and relative yield to evaluate the N efficiency of canola cultivars at the vegetative stage. Root morphological parameters including RL and RV may be associated with stem-leaf growth and N uptake on rice (*Zhang et al., 2009*). In this study, RL and RV were found to be highly correlated with dry matter weight and N accumulation. This may be because biomass and N accumulation increased with the development of roots (*Zhang et al., 2009*). N efficiency in maize was also found to be related to the plant root system's ability to extract available N from the soil profile (*Worku et al., 2012*). SNA and PNA as N accumulation parameters can directly reflect differences in N efficiency of alfalfa cultivars. *Xu, Fan & Miler (2012)* found that there were differences in N efficiency including total N accumulation, N uptake, and N translocation. Plant biomass, root morphology and N uptake parameters were used to evaluate the N efficiency of oilseed rape (*Wang et al., 2014*). *Chun et al. (2005)* showed that root growth, N uptake and yield could be used to screen different N efficiency in maize cultivars. Therefore, biomass, root parameters and N accumulation are often used as parameters for N efficiency evaluation.

## Alfalfa can be classified into four N efficiency types

In this study, we comprehensively analyzed the N efficiency types of 28 alfalfa cultivars by membership function and cluster analysis, using the parameters at seedling stage. This differed to studies that evaluated single or average performance of N efficiency of crops using similar N efficiency parameters (*Balint & Rengel, 2008*; *Chen et al., 2013*). Given N efficiency of crop is very complex, evaluation of crop N efficiency by a more comprehensive analysis may achieve more accurate results. Based on our analysis, 28 alfalfa cultivars were classified into four N efficiency types: Very efficient (LW6010 and Gannong NO.5), Efficient (Gannong NO.3 and Xinmu NO.1), Anti-efficient (Gannong NO.7), and Inefficient (Longdong). Different crops have different classifications of N efficiency types. For instance, maize was classified into three N efficiency types (Sensitive, Intermediate and Non-sensitive) by *Tsai et al. (1983)*. In this classification, the sensitivity could represent

the ability of crop responding to environmental N but not the N efficiency of crops. The classification of N efficiency of maize by *Kumar, Joshi & Dagla (2013)* was efficient and responsive, Non-efficient and Responsive, Non-efficient and non-responsive, and Efficient and non-responsive. *He et al. (2017)* divided oilseed rape cultivars into four N efficiency types: N-responder, N-non-responder, N-efficient and N-inefficient. Standards of these classifications were N-responder and N-efficient, which may be confused in the application sometimes. In our study, N-efficiency was the only classification standard. Maize was classified into four types by *Chen et al. (2013)*: Efficient-efficient, High-N efficient, Low-N efficient, Nonefficient-nonefficient. In *Chen et al. (2013)*'s classification, the relationship between environment N and crops was indicated clearly. In our classification, N efficiency types directly reflected N efficiency of alfalfa, which is intuitive and prominent. Besides cereal crops mentioned above, classification of N efficiency of harvesting vegetative crops also was reported. For example, *Liu et al. (2012)* classified into Efficient-efficient, High-N efficient, Low-N efficient, Nonefficient-nonefficient in spinach. According to N efficiency characteristics of cucumber under medium and low N levels, cucumber cultivars were classified into four types: Efficient-efficient, Efficient-inefficient, Inefficient-efficient, and Inefficient-inefficient by *Zhao et al. (2015)*. In our study, classification of N efficiency types directly reflected N efficiency of alfalfa. Thus it is a suitable N efficiency screening system of alfalfa, which we called "N efficiency seedling screening system". Different N efficiency types have different performance characteristics and their management and application in production are also different. We can apply N fertilizer according to the N efficiency types to achieve high herbage and protein yield, increase N fertilizer utilization rate, and reduce wastage (*Kant, Bi & Rothstein, 2011*). In our study, "Very efficient" alfalfa cultivars have stronger adaptability and higher forage utilization value. So, "Very efficiency" alfalfa cultivars such as LW6010 and Gannong NO.5 can be grown with less N fertilizer, which may still exhibit better yield and quality than other types. *Chen et al. (2013)* found that the average yield of "Very efficiency" maize cultivars was higher than the less N efficient cultivars. Under low N level, medium N level, and high N level, the yields of "Efficiency" cultivars were 15%, 6.62%, and 7.57% higher than the average yields of all tested cultivars, respectively, and the potential N fertilizer savings was 25.2–15.9%. *Worku et al. (2007)* reported that "Efficiency" maize cultivars had the potential for a 10.7% increase in yield and 12.7% reduction in N fertilizer input. N efficient maize cultivars could be suitable for high and low input farming systems, which would reduce cost of management in terms of nitrogenous fertilizer application (*Kumar, Joshi & Dagla, 2013*). It was indicated that the N efficient maize cultivars generally had considerably high yield potential under high N. Thus, these N efficient cultivars could be suitable for high and low input farming systems and would reduce cost of management in terms of nitrogenous fertilizers. According to the different performances of different N efficiency types of alfalfa on different N levels, the possible N fertilization measures with different N efficiency types of alfalfa in the field were suggested. In this study, "Efficient" alfalfa cultivars, *e.g.,* Gannong NO.3 and Gannong NO.4 were suggested to increase yield maybe through proper apply N fertilizer. Our previous study found that plant height, yield and quality of Gannong NO.3 were significantly higher than that of Longdong after N fertilizer application (*Liu et al., 2013*), indicating that

"Efficiency" alfalfa cultivars had high fertilizer return rate. *Chen et al. (2013)* showed that maize yield can be increased by 10%–15% and N fertilizer input reduced by 10%–20% if "Efficiency" cultivars were used in North and Northeast China. "Anti-efficient" alfalfa cultivars such as Gannong NO.7 and Gannong NO.9 have stronger tolerance to low N in soil than other types, so it could grow in land with less N addition. "Inefficient" alfalfa cultivars maybe perform poorly regardless of N fertilizer application. However, some of the "Inefficient" alfalfa cultivars have excellent performance in stress resistance. For example, Longdong has good salt and alkali resistance as well as cold resistance, so it can be used for ecological service objectives without pursuing high yield and quality (*Qi et al., 2015*). In short, the classification of N efficiency can not only explore N high efficiency alfalfa germplasm resources, but also guide the rational application of N fertilizer.

## The correctness of seedling screening was further proved through the whole growth period verification

There has been little report on whether N efficiency in crops at maturity can be reliably determined by screening germplasm in the vegetative stage. In order to validate the "N efficiency seedling screening system", we conducted verification studies during whole growth period. Overall, there was consistency in the N efficiency parameters (SDW, SNA and RL) between seedling stage and whole growth period for the four tested cultivars. N efficiency parameters were tested for consistency in N efficiency during the whole period. For example, SDW of "Very efficient" alfalfa (LW6010) was 50%, 46%, 47%, 35%, and 40% higher than that of "Inefficient" alfalfa (Longdong) at the seedling, budding, flowering, pod, and maturing stages, respectively, under the medium N treatment. This indicates that screening at seedling-stage could well predict what can happen at later stages up to maturity. SDW, RL and SNA of "Very efficient" alfalfa (LW6010) were higher than "Inefficient" alfalfa (Longdong) in each growth period. *Worku et al. (2012)* also found that an N-efficient cultivar (Hybrid 6) had greater root-length than an N-inefficient cultivar (Hybrid 7). Low N-tolerant maize selected in low N level often achieved a higher yield (*Presterl et al., 2003*). After the N concentration increases, SDW, RL and SNA of "Very efficient" (LW6010) and "Efficient" alfalfa (Gannong NO.3) obviously increased, while those of "Anti-efficient" (Gannong NO.7) and "Inefficient" alfalfa (Longdong) had relatively little increase. Thus, the classification of the N efficiency types in our study could probably be accurate, and "N efficiency seedling screening system" be used as an efficient screening tool for N efficiency of alfalfa cultivars.

## The correctness of seedling screening was further demonstrated through gene analysis

The reliability of "N efficiency seedling screening system" had also been confirmed by molecular method. Expressions of *NRT2* and *AMT1* play important roles in N uptake, plant growth and development under N stress conditions. In our study, different alfalfa cultivars had different expressions of *NRT2* and *AMT1* under medium and low N level, suggesting that N efficiency of cultivars may be different at the seedling stage. Expressions of *NRT2* and *AMT1* genes in "Very efficient" alfalfa (LW6010) were higher than that for the other three N efficiency types, suggesting that "Very efficient" alfalfa had strong

capacity in N uptake. This was also found in other crops, for example, the expression levels of *OsAMT1;1* and *OsNRT2;1* were higher in GD (seedling-stage high-NUE rice cultivar) compared to NG (seedling-stage low-NUE rice cultivar) (*Shi et al., 2010*). The expressions of *NRT* genes in "Efficiency" cultivars of oilseed rape were considerably higher than those in "Inefficiency" cultivars (*Wang et al., 2014*). The expressions of all of these genes were significantly higher in roots of the N-efficient genotype D4-15 than in roots of the N-inefficient genotype D1-1 under low N conditions. In this study, the changes of expressions of *NRT2* and *AMT1* genes in different N efficiency types were similar with the change of stem-leaf dry weight, whole plant dry weight, total root length, total root volume, stem-leaf N accumulation and whole plant N accumulation at seedling stage. There was consistency in the N efficiency between expressions of *NRT2* and *AMT1* genes and comprehensive value at seedling stage in four N efficiency types. Overall, these results could further confirm that "N efficiency seedling screening system" of alfalfa should be applicable. At the same time, *NRT2* and *AMT1* genes can be used to quickly screen out high N efficiency types and low N efficiency types of alfalfa.

## CONCLUSIONS

N efficiency screening system of alfalfa was established at seedling stage, which would accelerate the screening time. Alfalfa can be divided into four N efficiency types. The changes of evaluation parameters in different growth stages have the certain continuity and correlation among the different N efficiency types of alfalfa, that is, N efficiency screening of alfalfa can be carried out in each growth period. *NRT2* and *AMT1* genes can be used to quickly screen out high N efficiency types and low N efficiency types of alfalfa.

## ACKNOWLEDGEMENTS

The authors thank Weibiao Liao and Zhongnan Nie at Gansu Agricultural University for their suggestions and English correction. The authors also thank a reviewer for suggesting the principle component analysis.

### Funding
This work was supported by the National Natural Science Foundation of China (31460622) and special funds for discipline construction (GAU-XKJS-2018-008). The funders had no role in study design, data collection and analysis, decision to publish, or preparation of the manuscript.

### Grant Disclosures
The following grant information was disclosed by the authors:
National Natural Science Foundation of China: 31460622, GAU-XKJS-2018-008.

### Competing Interests
The authors declare there are no competing interests.

## Author Contributions

- Xiaojing Liu conceived and designed the experiments, authored or reviewed drafts of the paper, and approved the final draft.
- Yajiao Zhao analyzed the data, prepared figures and/or tables, authored or reviewed drafts of the paper, and approved the final draft.
- Feng Hao performed the experiments, prepared figures and/or tables, and approved the final draft.

## Data Availability

The original measurements are available in the Supplementary File.

## Supplemental Information

Supplemental information for this article can be found online at http://dx.doi.org/10.7717/peerj.13343#supplemental-information.

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
