# Peer review of "Development of nitrogen efficiency screening system in alfalfa (Medicago sativa L.) and analysis of alfalfa nitrogen efficiency types"

_PeerJ, doi:10.7717/peerj.13343_

## Round 0.1 · original submission · Major Revisions

All the reviewers recommend major revision. The text should be rewritten. There are issues with the English presentation. Please take more time for revision (no hard deadline for resubmission).

Reviewer 1 ·

Basic reporting

This paper focuses on the screening of higher and lower N efficiency cultivars of alfalfa from growth parameter and gene expression. N efficiency parameters including stem-leaf dry weight (SDW), plant dry weight (PDW), total root length (RL), etc., NRT2 and AMT1 genes were demonstrated to be the effective parameters for screening the higher N efficiency cultivars of alfalfa. Therefore, this paper has a certain scientific value and academic significance in screening different N efficiency cultivars of alfalfa, however, the language needs to be completely polished, especially the abstract content.

Experimental design

“Materials & methods” are described with sufficient detail, however, it should be modified, including “the low N level, 21 mg N·L-1”, “Nitrogen Content” (It should be described concisely), “Nitrogen Efficiency” (It should be easily understood), etc.

Validity of the findings

This paper is a scientific exploration based on cultivation and gene expression experimental data. Good experimental results have been obtained, such as 4 N efficiency types of alfalfa cultivars, NRT2 and AMT1 genes and so on. It's a comprehensive research report.
However,most of the experimental results are disorganized, and the experimental data provided in this paper can not support the experimental results completely. It is suggested that the experimental results should be reorganized , and the table and picture should be reconstructed.

1. Some tables and pictures should be reprogrammed.
Table 1 was too long to be put in the paper, and it should be put in supplemental file.
In table 5, the number corresponding to cultivar should be put in.
In Fig 1, the number should be replaced by the cultivar name, for instance, “15” should be replace by “LW6010”.
In Fig 2 and Fig 3, the note part possess too little information, and the letters at the top of the histogram bars should be noted.
2. In introduction, the description of Alfalfa should be described in more details. In discussion, the titles are too simple.
Only a few problems are listed here. Careful revisions are needed further.

Additional comments

The author did much works for the paper, however, it was seem that more wirting experiences and serious attitudes were needed for you.

Reviewer 2 ·

Basic reporting

This article discusses an indicator of the effectiveness (use or absorption) of alfalfa N. It has been shown that efficiency parameters N, including dry stem-leaf weight (SDW), dry plant weight (PDW), total root length (RL), etc., NRT2 and AMT1 genes are important markers for screening plants for the trait under study. ... The article is of interest to a wide range of geneticists and breeders of alfalfa. However, the language needs improvement - the text contains inconsistent sentences, typos and the sudden use of a capital letter at the beginning of a word. For example, the sentence on line 48 says "Arabidopsis", followed by "rice".

Experimental design

In the materials and methods, it is not entirely clear how the inoculation was carried out - and what specific microorganisms were used. However, this whole part is quite detailed and allows you to establish the scheme of the experiment. It is quite suitable for the formation of some conclusions.

Validity of the findings

Despite the large range of data obtained, they are poorly integrated and insufficiently discussed. the results are difficult to read due to weak English, enumerations and references to tables. Better to reorganize text and illustrations in an analytical way. For example - Figures 1 and 2 represent a set of the same sotra in different conditions - a comparison of these dendrograms suggests itself. in addition, such data can be presented on a two-dimensional graph - much more clearly. Column graphs in Figure 2 and, to some extent, by 3 can also be in the application, while in the text of the article something integrative would look good - profiles of the dynamics of development or analysis of principal components or principal coordinates, with the allocation of clearly distinguishable groups. infinite small columns reduce the quality of perception. in addition, if you consider them as individual comparisons, without integration, then you need to calculate the multiple comparison correction (FDR)

Additional comments

try to be stricter with statistics, but more creative with the presentation of the results

Reviewer 3 ·

Basic reporting

I suggest a thorough review of the English. For example, line 66, "NRT2 and AMT1 genes were high affinity genes." should be "NRT2 and AMT1 are high affinity transporters".

It is confusing when a lot of abbreviations are used. It is not common to use "N" instead of "nitrogen".

In the background, Authors spent a lot of effort to explain the difference between "N efficiency" and "N use efficiency". It is confusing and it is not necessary for the topic.

Authors explained the reason of using NRT2 and AMT1 as marker genes for NUE. Some references where people has done so are needed.

Not enough information about alfalfa. It is important to let reader know what is the yield of alfalfa. The ultimate goal should be less fertilizer, less waste with higher NUE and higher yield.

Experimental design

Authors defined NUE as shoot dry weight / total plant nitrogen accumulation, where the latter is defined as plant nitrogen content x plant dry weight. So the NUE, according to authors, is the shoot part (vs root in dry weight) of the total assimilated nitrogen. For me, authors are describing the ability of a plant to transfer assimilated nitrogen to the shoot and it is irrelevant to the ability of a plant to absorb and assimilate nitrogen from the soil or fertilizer, which will not help in guiding the fertilizer application in real practice.

Alfalfa is a legume, and its NUE is directly linked to its native rhizobium strain that the alfalfa is in symbiosis with, at least, the dominant strain. But authors used one single strain (without given any information on it) to inoculate all the alfalfa varieties in the study. On top of that, no information about nodulation is provided. So it is not clear how much nitrogen is directly absorbed by root, or from symbiosis.

An explanation why only NRT2 and AMT1 is selected as marker for NUE? On top of that, there should be multiple members of NRT2 and AMT1 family, what's the expression of the other members? Only one reference gene is selected, "18s" and no reference is provided for this gene. Also, no primer efficiency is provided.

Numbers and units should be carefully reviewed before submission. Line 104 "the low N level, 21 mg N/L", but later all the number referred to "N2.1". Line 181, "Five grams of total RNA"? or 5 ug?

Validity of the findings

The data is well analyzed. Figure legends need to be more specific. Information like repeat number and statistic analysis used should be added to legends, so that readers can interpret the data without influenced by text.

My question about the definition of NUE and its relevance to the real agricultural practice remains. So I cannot comment on the impact of this research and I do not know how much can be translated into the real field.

Additional comments

This manuscript is not well prepared. Issues with the English, inconsistency of numbers and units should be carefully resolved before submission.

---

## Round 0.2 · Major Revisions

Both the reviewers have critical remarks demanding major revision. Please check the abstract (cultivars naming) and English presentation.
The manuscript still needs to be rewritten.

Reviewer 1 ·

Basic reporting

Even though you have revised the language of this paper completely, we still consider that the language should be polished using language editing services of PeerJ.
From the abstract, we can not see the reasons why four N efficiency types of alfalfa cultivars are classified: (1) Very efficient, e.g. LW6010; (2) Efficient, e.g. Gannong No.3; (3) Anti-efficient, e.g. Gannong No.7; and (4) Inefficient, e.g. In the abstract, N2.1 should be replaced by N21. The abstract content needs to be rewritten.

Experimental design

The root scaning content in Morphological Parameters part should be described in detail. Careful revisions are needed further.

Validity of the findings

In the paper, nitrogen (N) should be indicated for the first time, then nitrogen should replaced by N. “P” should be replaced by “p”.
1. Some tables and pictures should be reprogrammed.
In table 4, the ANOVA method should be used to indicated the significant differences among cultivars in N efficiency.
Fig 2 and Fig 3 should be improved by sigmaplot or origin software. There were some mistakes for letters at the top of the histogram bars in Fig 2 and Fig 3.
2. In discussion, the first title should be deleted. The other titles should be modified.
Only a few problems are listed here. Careful revisions are needed further.

Additional comments

The quality of the paper is far from meeting the demands of publishing on <PeerJ>. It is best to consult your tutor more in revising this paper.

Reviewer 2 ·

Basic reporting

The work is devoted to the important problem of assessing the efficiency of nitrogen use by alfalfa plants. The proposed method can make genetic experimentation and selection more intense.

The article provides the rationale and goals for the development of such a method, briefly but sufficiently describes the background.

The work is written in understandable language, but full of stylistic flaws. It is worth paying attention to the use of the words "phenomenon", "development" in all cases of their use. There are proposals that seem inconsistent. So, for example, on lines 79-81 the meaning eludes the reader.

The experimental part is described in sufficient detail, however, on line 139 it is necessary to clarify that you are measuring gene expression. Otherwise, the reader may misunderstand you. I would also recommend briefly describing the parameter estimation method you are referring to on line 148.

An obvious drawback of the work is the poor description of the statistical analysis. It is necessary to add details to the description and provide the motivation for using the methods, in particular the membership function. There is also a related error when doing many pairwise comparisons. And you need to either add a correction for multiple comparisons (bonferonni or one of the benjamini-XX, depending on the task), or justify why you do not make such an amendment. The text mentions various statistical estimates, for example, the coefficient of variation, all of them are also worth mentioning in the materials and methods.

Figure 2, which contains a lot of bar graphs, can be presented in the appendix, while in the text of the article, a principal component or principal coordinate analysis would look good. Genotypes should be used as objects on the scatter plot, and features can be displayed as projections (biplot). Thus, Figure 2 can be replaced with 3 scatter plots, and maybe even the common one.

Experimental design

does not cause big remarks

Validity of the findings

in addition to the already mentioned correction for multiple comparisons, the cluster analysis shown in Figure 1 is not described in the materials and methods. To be honest, the allocation of clusters raises questions. Due to the fact that 4 types of efficiency of nitrogen assimilation are given in the discussion, it is necessary to describe in detail why the clusters were identified in this way and not otherwise.

---

## Round 0.3 · Minor Revisions

The manuscript still needs revision. Please check the comments in the annotated file provided by the reviewer. Please update the text presentation, and order of the sections in the manuscript.

Reviewer 1 ·

Basic reporting

Corresponding changes has been made, however, there are still a lot of mistakes that need to be corrected. For instance, N2.1 should be replaced by N21 in abstract.

Experimental design

no comment

Validity of the findings

1.In the paper, “nitrogen (N)” should be indicated for the first time, and then nitrogen should replace by N.
2.The P should be replace by p.(line 212)
3.Three paragraphs in conclusion part should be combined into one.
4.In Fig3, Ⅰ, Ⅱ, Ⅲ, Ⅳ and Ⅴ should be indicated by corresponding growth periods.
5.Fig3 and Fig 4 should be improved further, and the letters at the top of the histogram bars should be indicated by related information.
6.The mistake for the letter in Fig 4 has been marked by red line.
Only a few problems are listed here, and careful revisions are needed further.

Additional comments

More serious attitudes are needed in revising this paper.

Annotated reviews are not available for download in order to protect the identity of reviewers who chose to remain anonymous.

---

## Round 0.4 · Minor Revisions

Excuse me for the delay in the manuscript processing. We have received contradictory reviewing remarks. One of the reviewers suggests rejecting the manuscript, demanding additional field experiments. I think the discussion could be extended based on existing experiments. Please, tone down the statements in the conclusion. I look forward to your re-submission of the updated manuscript.

Reviewer 1 ·

Basic reporting

Corresponding changes has been made, however, there are still some mistakes that need to be corrected, especially in table and figure parts.

Experimental design

No comment

Validity of the findings

1. In table 4, the details of the ANOVA analysis should be written in “Note” part.
2. In Fig 3 and Fig 4, some markings such as cultivars, growth periods, N levels and organs should not be recurring, which spoiled the beauty of figures.
3. Fig3 and Fig 4 should be improved further.

Additional comments

Corresponding changes has been made, however, there are still some mistakes that need to be corrected, especially in table and figure parts.

Reviewer 3 ·

Basic reporting

N/A

Experimental design

N/A

Validity of the findings

Authors concluded that they have classified alfalfa into four different N efficiency types and suggested that there should be different managements and applications of N in alfalfa production. However, no experiment was done in a real production condition (with both experiments in sand filled pots). And the most serious default is that authors didn't consider the N symbiosis between alfalfa and rhizobium. Legume crops, unlike cereals, they have the ability to fix N through N symbiosis. In the real production, the rhizobium in the field is surely more complicated than single Sinorhizobium meliloti inoculation used in this experiment. Authors profiled the gene expression of NRT2 and AMT1 genes, but they are more relevant to nitrogen use efficiency in cereal crops. NUE is a quantitative trait, where multi-year large scale field trial is needed for selecting high NUE variety. Therefore, I do not see much value in this research without at least three cycles of field trial.

---

## Round 0.5 · Minor Revisions

Thanks for the manuscript updates and revisions. There are no critical remarks now, only suggestions for the figures quality. Please update it as suggested by reviewer #1.

Additionally the Section Editor has commented and said:
"The abstract needs to be rewritten so that it presents an understandable summary of the research summary. For example, instead of using "N21" and "N210" in the abstract use low and medium nitrogen (or define N21 and N210). It is unclear what a multivariate scores is and what it means for it to be less or greater than 0.5. Line 15 " have the higher coefficient of variation and correlation" higher than what? Correlation to what?

line 43, Arabidopsis is not a crop.

The statistical methods for qPCR normalization and analysis need to be provided."

I think this will not need an additional reviewing round. I am making a decision of Minor Revisions now assuming all the remarks will be taken into account.

Waiting the final version submission.

Reviewer 1 ·

Basic reporting

Corresponding changes have been made, however, figure 3 and 4 need to be improved.

Experimental design

No comment

Validity of the findings

1.Figure 3 and 4 should be improved on the basis of the previous manuscript. The aesthetics and standardization of these pictures are too bad.
2. The minor tick marks on the ordinate in figure 3 and 4 should be deleted.

Additional comments

No comment

Reviewer 2 ·

Basic reporting

The article has been substantially revised and can be published without significant changes.

Experimental design

Adequate

Validity of the findings

Adequate

Reviewer 4 ·

Basic reporting

Accept in present form

Experimental design

Accept in present form

Validity of the findings

All underlying data have been provided

---

## Round 0.6 · Minor Revisions

Sorry for multiple reviewing rounds, but there are some remarks demanding revision. Please update the figures accordingly.

Reviewer 1 ·

Basic reporting

Corresponding changes have been made, however, figure 3 and 4 should be improved.

Experimental design

No comment

Validity of the findings

Figure 3 and 4 should be improved on the basis of the previous manuscript (reviewing-58454-v3). The ordinate values of Figure 3 and 4 should be all on the left.

Additional comments

No comment

---

## Round 0.7 · Minor Revisions

I think the manuscript passed enough revision rounds and сould be accepted now. The reviewers have no more critical remarks.

The figures have appropriate quality now.

However, I have some technical remarks to fix demanding some revision.

First is formatting. Check sign near the affiliation of the corresponding author - ‘Liu฀,’.
Email address looks strange - corresponding_author_liuxj@gsau.edu.cn -
Is it correct or should be ‘corresponding author liuxj@gsau.edu.cn’ ?
(change underscore sign ‘_’ to space ‘ ’)
Some phrases in English should be rewritten.
Standard term is ‘nitrogen use efficiency in crops’, not just ‘nitrogen efficiency’
In the Abstract please rewrite
Line 9: ‘Screening high nitrogen (N) efficiency crops...’ - change to ‘Screening nitrogen (N) use efficiency in crops...’
Overall, the Abstract should be rewritten. Need list the cultivars used (Gannong NO.3 and so on), avoid abbreviations like ‘LW6010 etc.’ - what is ‘etc.’ here?
Need check English again.

The phrase ‘The results showed that SDW, plant dry weight (PDW), RL, total root volume (RV), stem-leaf N accumulation (SNA) and PNA showed significant positive correlation both, and the coefficient of variations of SDW, PDW, RL, RV, SNA and PNA were all more than 20% under low N level (21 mg•L –1 N, designated N21) and medium N level (210 18 mg•L –1 N, designated N210) .’ - is too long, and not clear. Also it is not correct in English.
Fix it as two sentences, not use wording ‘The results showed’, remove word ‘both’. Remove extra numbers like ‘21 mg•L –1 N,’, it could be defined in the main text.

The concluding phrase in the Abstract should be rewritten ‘Four N efficiency types of alfalfa cultivars were classified:...’ - change to more precise like ‘We developed new classification system of N efficiency types of alfalfa cultivars. It proved its effectiveness on 28 alfalfa in China.’ (In the main text I see Table with only 14 cultivars)
28 alfalfa cultivars should be briefly described in the text. Why it is important to study N efficiency on these cultivars, not just on other crop plants? (Line 84)

The formulas in the text (lines 160-162) should be formatted as formula, from separate lines, with proper comments. Use Italic font for parameters.

The journal has wide readers’ circle. The text should be understandable for readers.

The Results section should be commented by more detailed phrases -
Line 218 - ‘significant positive correlations’ - what does it mean for classification system?
Line 224: ‘The coefficient of variations (CVs) of SDW, RDW, PDW, RL, RV, SNA, RNA and PNA were all more than 20%’ - what it means for classification system? May use any parameter?

Use wording ‘Under N21 condition’ or ‘At low N supply’, not just ‘under N21’.

The Note under the Tables may not repeat:
“Note: SDW, RDW, PDW, SH, RL, RV, SNC, RNC, PNC, SNA, RNA and PNA represent stem-leaf dry weight, root dry weight, whole plant dry weigh, plant height, total root length, total root volume, stem-leaf nitrogen content, root nitrogen content, whole plant nitrogen content,
stem-leaf nitrogen accumulation, root nitrogen accumulation and whole plant nitrogen
accumulation.”
Add word ‘correspondingly’ after the sentence. Or rewrite to show each abbreviation in full like
“Note:
SDW - stem-leaf dry weight;
RDW - root dry weight;
Use this note for Table 1, and refer to it in next tables as ‘(see the note to Table 1)’.
To make reading more convenient.

Some references in Chinese like ‘Chun L, Chen FJ, Zhang FS, Mi GH. 2005.’ have long links. It is better fix DOI numbers only, check the references formatting.

Figure 1 has low resolution. Please present the figure in higher resolution, more sharp.

Overall, I recommend checking the English again.

---

## Round 0.8 · accepted · Accept

Thank you for the manuscript updates and rebuttal letter. The reviewers had no critical remarks. I think this work have to published now. My recommendation is to accept.

Please check English again, especially in the recently revised text.

Please add publication data (page numbers) to the reference 'Kou JL. 2011. Effect of nitrogen forms ... (in Chinese)'